# Data Augmentation Using Background Replacement for Automated Sorting of Littered Waste

**DOI:** 10.3390/jimaging7080144

**Published:** 2021-08-12

**Authors:** Arianna Patrizi, Giorgio Gambosi, Fabio Massimo Zanzotto

**Affiliations:** Dipartimento di Ingegneria dell’Impresa, University of Rome Tor Vergata, I-00133 Rome, Italy; giorgio.gambosi@uniroma2.it (G.G.); fabio.massimo.zanzotto@uniroma2.it (F.M.Z.)

**Keywords:** automated waste sorting, convolutional neural networks, background replacement, data augmentation, computer vision, deep learning, multi-class classification

## Abstract

The introduction of sophisticated waste treatment plants is making the process of trash sorting and recycling more and more effective and eco-friendly. Studies on Automated Waste Sorting (AWS) are greatly contributing to making the whole recycling process more efficient. However, a relevant issue, which remains unsolved, is how to deal with the large amount of waste that is littered in the environment instead of being collected properly. In this paper, we introduce BackRep: a method for building waste recognizers that can be used for identifying and sorting littered waste directly where it is found. BackRep consists of a data-augmentation procedure, which expands existing datasets by cropping solid waste in images taken on a uniform (white) background and superimposing it on more realistic backgrounds. For our purpose, realistic backgrounds are those representing places where solid waste is usually littered. To experiment with our data-augmentation procedure, we produced a new dataset in realistic settings. We observed that waste recognizers trained on augmented data actually outperform those trained on existing datasets. Hence, our data-augmentation procedure seems a viable approach to support the development of waste recognizers for urban and wild environments.

## 1. Introduction

Even though sorting and recycling efficiencies are improved by augmenting modern waste treatment plants, another problem is arising. More and more waste is being littered in the environment instead of being collected properly. Oceans are full of plastic debris [1,2], which is spoiling the beauty of coasts [3]. Yet, even if this is a huge problem for oceans, it becomes gigantic for land. It seems that the presence of macro-plastic debris in soil is 40 times bigger than in the oceans [4]. Pieces of solid waste are commonly found in streets, urban parks, beaches, forests and many other places, showing the footprint of a modern human society. Precious glass bottles, plastic bottles, cans and carton boxes are littered, and their potential value is lost. Moreover, the COVID-19 pandemic has made waste management even more challenging, due to uncontrolled increase of household waste during lockdown [5].

The solution to retrieve macro-litter cannot just rely on the current trend of augmenting traditional plants for management of solid waste with Automated Waste Sorting systems based on image recognition techniques. Actually, litter has bypassed traditional recycle management, which can transform waste in new precious material. Current waste recognition systems aim to reduce the number of people working in these unhealthy waste treatment plants. For example, waste pickers of the most important landfills in Brazil have been, or still are, subject to many diseases, due to working conditions: mainly osteo-muscular disorders, but also viral infections and emotional vulnerabilities [6]. Therefore, current waste recognition systems [7,8,9,10,11,12,13,14,15,16,17,18,19,20,21,22] as well as current corpora for training [7,10] focus on the activity of waste sorting in waste management plants. Solid waste runs on conveyor belts and, then, it is sorted into the different types. For this reason, current datasets contain waste images taken on a uniform background. In fact, conveyor belts have generally a uniform color. Yet, systems resulting from these datasets can be used only when solid waste has arrived in its prefinal destination. Searching and sorting solid waste in natural and urban environments may result in several difficulties for these systems trained on existing datasets.

In this paper, we introduce a method for building automated waste recognizers that can be used for identifying and sorting litter directly where it is found. Our method is based on BackRep, a novel data-augmentation procedure, which expands existing datasets by cropping solid waste in images taken on a uniform (white) background and superimposing it on more realistic backgrounds. For our purpose, realistic backgrounds are those backgrounds representing places where solid waste is usually littered. To experiment with our data-augmentation procedure, we produced a new dataset in realistic settings.

## 2. Related Work

Automated Waste Sorting (AWS) has been sometimes framed as an image recognition task [12] and, indeed, this could be an interesting way to see it. However, this task poses additional challenges to the classical object recognition in images. Actually, unlike normal objects, waste objects are distorted, possibly crushed. From our perspective, AWS should be seen as an image recognition task, as we envisage to build robots operating in different scenarios to retrieve littered waste.

The major challenge to design AWS systems learned from data is building an image dataset for the specific task. Hence, this has been one of the first issues that has been tackled along with the design of AWS systems. One of the most important datasets is the one gathered during the development of TrashNet [7]. This dataset has been the basis for many other studies [8,9,10,11,12,13,14,15,16,17,18,19,20,21,22]. The TrashNet dataset has been further extended by adding the compost class [10]. However, this type of dataset is important but not completely useful, as waste images are taken on a neutral white background, which is probably used to simulate conveyor belts, i.e., the operational environment of AWSs. Here, we propose a novel testing dataset to simulate a different operational scenario for AWSs.

As the aim of AWS systems is seen as an object recognition task in images, applying deep-learning architectures is one of the most examined approaches. Convolutional Neural Network (CNN)-based models such as AlexNet [23] have been used for building solutions such as TrashNet [7]. Yet, different models have been proposed by increasing the depth and even modifying the structure of the neural network. In this framework, RecycleNet [8] improved dramatically the performance obtained on the TrashNet dataset. RecycleNet extends a 121 layers DenseNet [22] with the use of skip connections via concatenation on Models with many layers combined with application of transfer learning.

As the expected use of AWS systems is in apps in mobile phones [10,14] or in software embedded in waste picker robots [21], existing datasets [7,10] and, consequently, existing learned AWSs are no more adequate. Waste picker robots may be placed in urban as well as natural environments to clean and recycle precious waste using adapted AWS systems.

Datasets representing waste where it is usually found are starting to be hypothesized [12], and a project to build a crowdsourced multilabel dataset is in progress [24]. However, since, similar wastes may appear on quite a lot of different backgrounds and conditions, gathering a reliable dataset for the AWS task is really challenging. For this reason, we believe the data augmenting approach proposed here may greatly contribute to generalize the classification model by exploiting conveyor-belt-oriented datasets to our advantage.

The use of data-augmentation techniques is quite common in the framework of image classification, usually applied to increase both the amount and the diversity of data (images), reducing the dependence of the classification from image characteristics which are assumed irrelevant, such as scale, brightness, rotation, filtering, partial erasing. Augmentation is then performed by applying (sequences of) operators, which are assumed to represent invariances with respect to classifications, on the images. Given a set of operators, finding an effective data-augmentation technique based on suitable compositions of such operators may greatly improve the overall classifier predictions. Although in many cases the proposed augmentation techniques are dataset-dependent, several techniques have also been proposed in the literature to learn an effective augmentation technique from the dataset itself, by searching a space of possible augmentation procedures [25,26,27,28]. In this paper, however, we concentrate on the case of data augmentation with respect to a different characteristic, i.e., image background. As observed above, this seems particularly relevant in the application framework considered. Litter may be found and should be classified in quite different contexts, in a way which must be as robust as possible with respect to such contexts, and thus to the image background.

The BackRep technique presented here is based on augmenting the dataset by inserting trash items from images taken on a uniform background on a set of different realistic backgrounds. It is an open issue to perform a detailed study of how much the choice of different backgrounds (in a given set) affects the classifier performance. We also observe that the approach of [25,26,27,28] to automatically derive augmentation technique from data seems suitable to be applied also in the framework considered here.

## 3. Datasets

To test waste recognizers in normal setting, we relied on an existing dataset– CompostNet [10]—which has been mainly used for training (Section 3.1) and we gathered a testing dataset where waste images are taken where waste can be most often found (Section 3.2).

### 3.1. An Existing Conveyor-Belt-Oriented Dataset

We referred to the CompostNet dataset [10] to test the ability of our data-augmentation method to exploit existing datasets for training waste classifiers that can be applied to identify and classify waste in a general setting.

In the CompostNet dataset, waste images are taken on a white background (Figure 1). For this reason, these images are oriented to support the building of waste classification models which can be used for sorting waste on conveyor belts. In fact, the CompostNet dataset completes the TrashNet dataset [7] adding the *compost* class. The TrashNet dataset consists of 2527 images on six different classes of materials: *paper, glass, plastic, metal, cardboard* and *trash* (see Table 1). These images represent ruined objects, which can be associated with the notion of waste. These images were produced as part of the activities reported in [7] and they were taken from multiple viewpoints to expand (and improve) the dataset with additional images representing the complex shapes of damaged waste. Their initial idea of using the Flickr Material Database [29] and images from Google Images^TM^ was not applicable. In fact, these images did not accurately represent the state of wasted goods as most of them represents undamaged or totally unused objects. The TrashNet dataset has been largely used [7,8,10,11,13,15,16,20,21]. The set of additional images in CompostNet has the same features of the rest of the TrashNet dataset. The 177 compost images have a white background (see Table 1).

In our experiments, we used the 70%-13%-17% split for train, validation and test, respectively, as in [7]. All the categories have been included. Validation split will be useful for tuning hyperparameters during the learning phase. Hence, the model will be tested on these samples so that the gold test set will remain unused.

### 3.2. Littered Waste Testset

For evaluating whether waste recognition systems can provide good results for waste in normal setting, we prepared a testing corpus: the Littered waste Testset. In our corpus, images of waste items are taken on more normal settings, i.e., the background is not a neutral white. This aims to represent waste where it is most usually littered and not on the conveyor belt of a waste management plant.

The Littered waste Testset consists of 114 images categorized according to the categories of the CompostNet dataset (see Table 2). A group of 4 volunteers took the pictures in domestic and outdoor contexts with different kinds of backgrounds and light (see Figure 2). The group used their phone camera and images were then resized to 500 × 400 pixels. The distribution of classes of waste has been determined by the volunteers. These are waste found in urban environment and in their homes. Volunteers were asked to produce at least ten samples per class.

This novel testset may be confusing for waste categorization systems trained on conveyor-belt-oriented dataset. Actually, parts of the background may improperly affect the behavior of the convolution neural network, since the texture as well as the color may be similar to other kinds of waste. For example. the background of the plastic cap in Figure 2 may be mistaken for a carton. This makes the use of this dataset extremely challenging.

## 4. Methods and Implementation

The main goal of our research is building Automated Waste Sorting systems to identify and classify waste retrieved in its littered site, as represented in the Littered waste testset (Section 3.2). Hence, we first introduce here BackRep, our data-augmentation method for exploiting existing categorized waste datasets (Section 4.1). Secondly, we briefly report on AlexNet [7] and InceptionV4 [30], which are two state-of-the-art systems applied to waste classification. These two latter systems have been used in combination with BackRep to prove its viability with respect to the main goal.

### 4.1. Data-Augmentation with Background Replacement

The core of our method is a data-augmentation procedure based on image background replacement (BackRep). The intuition behind this approach is that of putting existing, categorized waste in context. BackRep is adding the noise that waste classifiers may find in Littered waste. Then, images in context help waste classifiers to focus on the most relevant parts of images.

The BackRep procedure relies on an existing conveyor-belt-oriented annotated dataset and consists of the following three modules (see Figure 3):

*Select&Crop* selects the waste from an image in a conveyor-belt-oriented dataset, crops it and, then, removes the background, replacing it with transparency. This is possible as images in conveyor-belt-oriented datasets are generally on a uniform background. Hence, selecting and cropping waste may be done with a high level of quality by means of existing, widely available image processing libraries. In our study, we used the OpenCV Python library.*Littered waste Background Selection* is a pseudo-random selection function of possible backgrounds. This pseudo-random function extracts backgrounds from available repositories. In our study, backgrounds derive from two major sources: (1) license free images found on Unsplash (https://unsplash.com, accessed on 5 August 2021); (2) background pictures produced in the present study. These backgrounds are randomly selected among pictures representing surfaces with different textures and lighting, as wastes can be found anywhere (see Figure 4).*Merge* is the simpler module as it merges cropped images and new backgrounds. The final output is a novel image with its trash classification label (see Figure 5).

### 4.2. Two Automated Waste Sorting Systems

To experiment with our main idea, we selected two different state-of-the-art models: AlexNet [23] as in TrashNet [7] and InceptionV4 [30]. In the following, we briefly describe the models and their experimental configuration.

Our version of AlexNet [23] is a sequential model slightly modified in line with TrashNet [7], which reduced the number of filters in the convolutional layers. In our experiments, we adopted the following additional modules and hyperparameters. First, VGG16 [31] image preprocessing module (from the KERAS library [32]) normalized images, even if AlexNet does not foresee the normalization of the input. This preprocessing module has been selected because it improved performance on the validation set and reduced the swinging shape of the loss curve, which was observed in [7]. Secondly, hyperparameters have been manually selected after some fine tuning over the validation set: the optimizer is Adam [33] with learning rate of 0.001 and models were trained for 20 epochs with batches’ size of 32 images. Model summary is shown in Table 3.

InceptionV4 [30] is a pre-trained architecture for image recognition applied to waste classification. One of the main ideas in this architecture is transfer learning. In fact, the model is pre-trained with ImageNet weights. Moreover, this architecture includes skip connections. In our experiments, basically, all the pre-trained weights have been fixed during training except for the last layer. We replaced the dense top layer with a network of a Global Average Pooling 2D layer with 1536 neurons and a fully connected layer with 1024 neurons and ReLU activation. This subnetwork aims to maximize feature extraction. The final classifier is a Dense layer with SoftMax function for output normalization with 7 neurons, one for each class. The model went through 80 epochs of training with mini-batches of 32 images. Additionally in this case, Adam [33] with learning rate of 0.001 has been used as optimizer.

## 5. Results

Experiments aim to investigate if our technique of Background Replacement (BackRep) for data augmentation (see Section 4.1) helps in adapting conveyor-belt-oriented Automated Waste Sorting (AWS) systems to recognize waste in realistic settings.

For the above reason, we experimented with two state-of-the-art systems–AlexNet [7] and InceptionV4 [30] (see Section 4.2)—enhanced with our BackRep. We texted four different configurations of the two systems: NotAug, BackRep, BackRep&NotAug and LittleAug. NotAug is the normal system. BackRep is the system where input data are augmented with our BackRep methodology. BackRep&NotAug is the combination of NotAug and BackRep, where input data is given as plain and augmented. Finally, LittleAug is the model fed with a dataset with little augmentation: images are presented in two ways: (1) plain; (2) vertically and horizontally flipped. This model is used to compare with baseline augmentation techniques.

We carried out the experiments on both datasets: conveyor-belt-oriented and Littered waste datasets (see Section 3). Section 3.1 describes the split in training, validation and test. Systems are trained on the conveyor-belt-oriented using the 4 configurations for 5 times and tested on conveyor-belt-oriented and Littered waste. We reported Accuracy, Macro Average F1 and Micro Average F1 (see Table 4 and Table 5). F1-score is a useful metric to understand classification performance because it depends on both Recall and Precision, and the final score is more influenced by lower values with respect to the arithmetic mean. The difference between Macro and Micro Average F1 is that the first one is the average of the unweighted F1-score per class, while the second one is the average of support-weighted F1-scores. Support is the number of samples for each class. All experiments were performed on Google’s free GPU on Google Collaboratory.

## 6. Discussion

Our experiments have solid bases as state-of-the-art waste classification systems are correctly used. In fact, the performances of both AlexNet and InceptionV4 are similar, on average, to those provided in the original papers [7,30]. Results are obtained as average of 5 runs with different seeds.

InceptionV4 has better performance on the conveyor-belt-oriented test with respect to AlexNet. In fact, AlexNet reached a 0.683(±0.020) accuracy whereas InceptionV4 reached 0.844(±0.009) (Table 4 and Table 5). The complexity of InceptionV4 is then well justified as the resulting performance wrt accuracy is about 16% higher.

However, our hypothesis is confirmed: conveyor-belt-oriented-trained systems are not directly applicable to Littered waste scenarios. Actually, both AlexNet and InceptionV4 had a consistent performance drop when tested in Littered waste setting. AlexNet dropped by more than 0.55 points in accuracy from 0.683(±0.020) in conveyor-belt-oriented test to 0.138±0.037 in Littered waste test (Table 4). The better InceptionV4 had a similar performance drop–around 0.44 in accuracy–from 0.844(±0.009) to 0.409(±0.012) (Table 5).

Our BackRep model is a viable solution to exploit conveyor-belt-oriented datasets for improving waste classification systems in the Littered waste scenario. The use of BackRep increases results in Littered waste setting in both systems. The improvement in accuracy is higher for the best system, which is InceptionV4: 0.442(±0.015) vs. 0.409(±0.012) in the Littered waste setting and this difference in accuracy is statistically significant (Table 5). Moreover, BackRep in combination with the normal dataset has even better results: 0.488(±0.025) vs. 0.442(±0.015). This is a clear and important improvement with respect to InceptionV4 learned on NotAug. Finally, the improvement is not only due to the increase in dataset size. In fact, InceptionV4 trained on BackRep&NotAug significantly outperforms the baseline data-augmentation model LittleAug.

Clearly, BackRep is not perfect. Actually, waste images may be misinterpreted and, then, wrongly cropped. For example, the can in Figure 5 in the Metal category can be barely recognized. Yet, this image used during training can help the neural network to focus on specific and important features of the can itself. Anyone can easily image a metal can littered in a field, where it can be covered of dirt or grass, so we are possibly training our network to recognize it even in those cases. The remaining wedge-of-moon-shaped can may be more informative than the complete can.

Our BackRep procedure as well as our Littered waste has a limitation. A large fraction of the littered waste is represented by micro-plastics [34]. In fact, micro pieces of plastic are not detectable with our current solution and these pieces may be so small they require different approaches, not based on image recognition techniques.

## 7. Conclusions

Littered waste is overwhelming. In this paper, we presented BackRep, namely a model to adapt existing Automated Waste Sorting (AWS) systems to be used for littered waste. We also proposed a novel dataset for testing AWS systems for littered waste. Experiments showed that AWS systems trained on augmented data outperform waste recognizers trained on existing datasets. Hence, our data-augmentation procedure BackRep provides a viable solution to build waste recognizers for urban and wild environments.

In conclusion, our BackRep model may enable the construction of Automated Waste Sorting systems in Littered waste settings. These systems can be used to equip swarms of robots, which may be used to clean woods, forests, shores and urban environments removing macro littered waste.

Clearly, our BackRep model is a general data-augmentation model, which can be used for different image recognition tasks. Hence, this model opens an interesting line of research.

## Figures and Tables

**Figure 1 jimaging-07-00144-f001:**
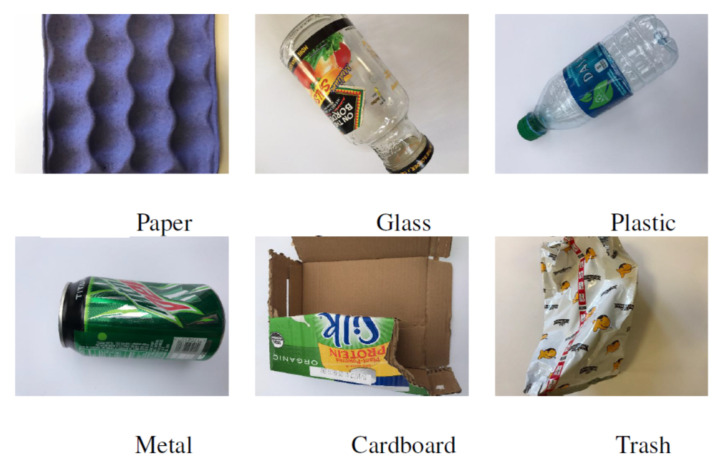
Trashnet dataset, one sample for each class.

**Figure 2 jimaging-07-00144-f002:**
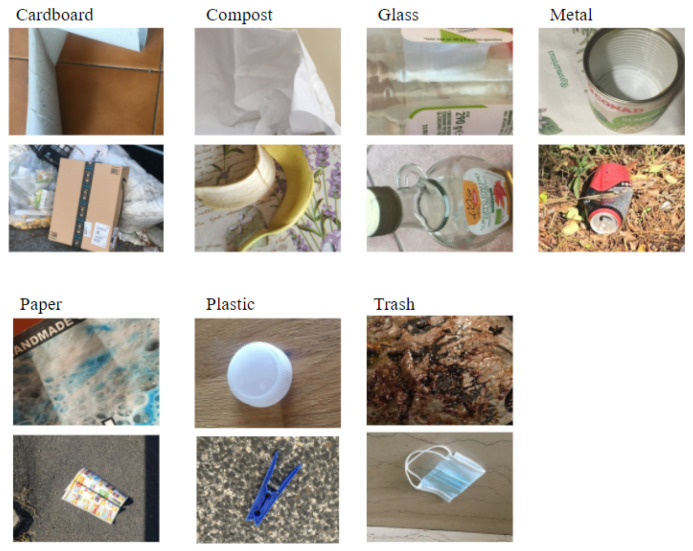
Littered waste testset: two samples for each class.

**Figure 3 jimaging-07-00144-f003:**
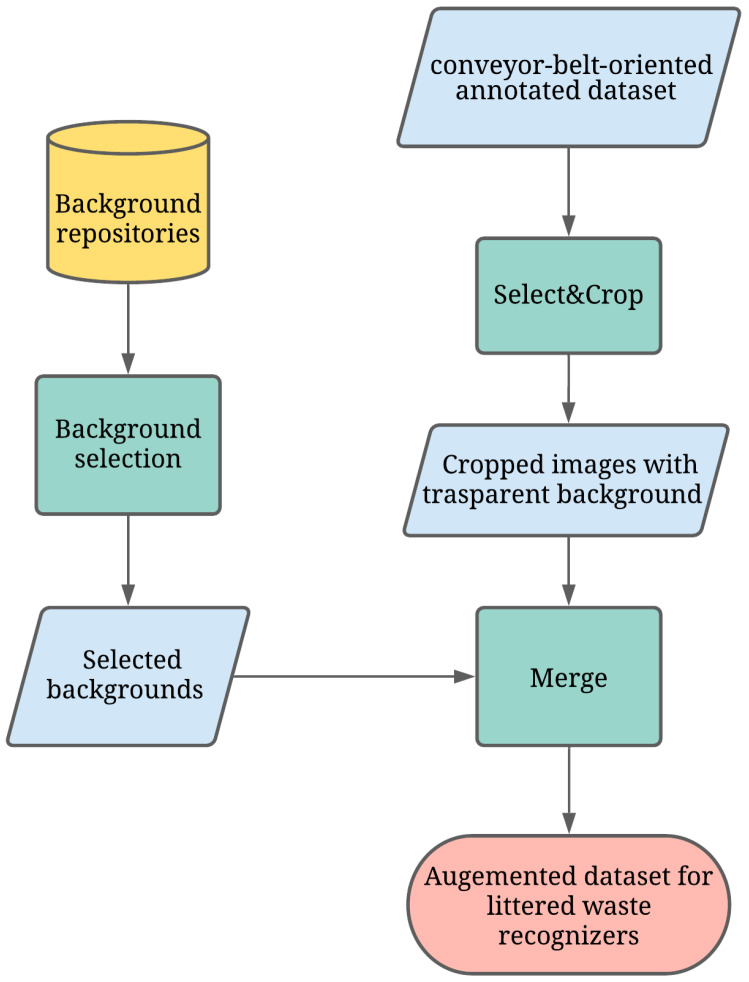
BackRep: a data-augmentation procedure based on image background replacement.

**Figure 4 jimaging-07-00144-f004:**
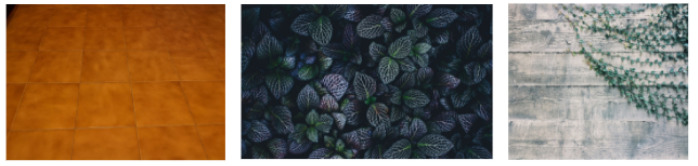
Background samples.

**Figure 5 jimaging-07-00144-f005:**
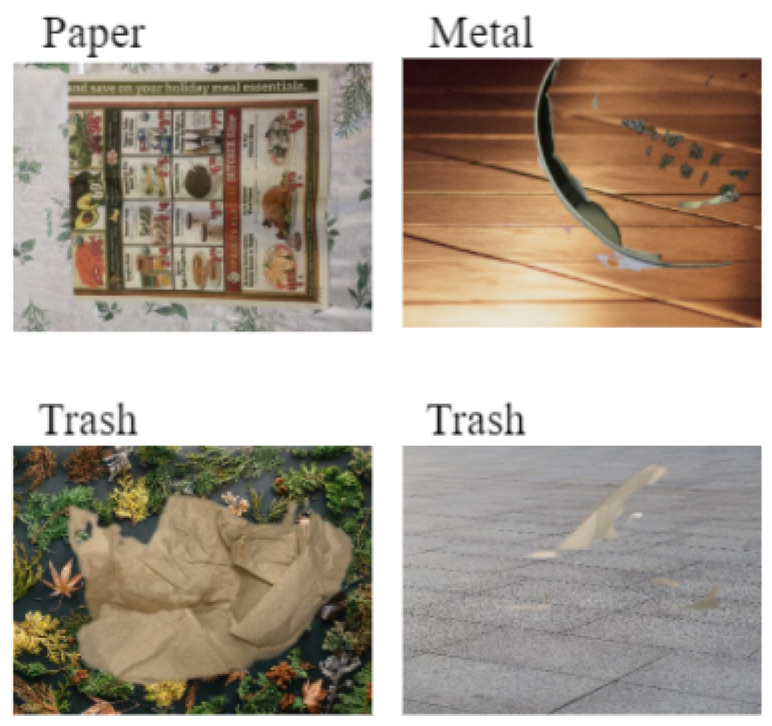
Overlay samples.

**Table 1 jimaging-07-00144-t001:** TrashNet and CompostNet Dataset distribution.

	Category	Entries	Percentage
TrashNet Categories	Cardboard	403	15
	Glass	501	18
	Metal	410	15
	Paper	594	22
	Plastic	482	17
	Trash	184	7
	Compost	177	6
	Total	2751	100

**Table 2 jimaging-07-00144-t002:** Distribution of the Categories on the Littered waste Testset.

	Category	Entries	Percentage
TrashNet Categories	Cardboard	15	13
	Glass	11	10
	Metal	11	10
	Paper	15	13
	Plastic	26	23
	Trash	16	14
	Compost	19	17
	Total	114	100

**Table 3 jimaging-07-00144-t003:** AlexNet parameter summary.

Layer (Type)	Output Shape	Param
Conv2D	(None, 64, 64, 96)	34,944
MaxPooling2D	(None, 32, 32, 96)	0
Conv2D	(None, 32, 32, 192)	460,992
MaxPooling2D	(None, 16, 16, 192)	0
Conv2D	(None, 16, 16, 288)	497,952
Conv2D	(None, 16, 16, 288)	746,784
Conv2D	(None, 16, 16, 192)	497,856
MaxPooling2D	(None, 7, 7, 192)	0
Flatten	(None, 9408)	0
Dense	(None, 4096)	38,539,264
Dense	(None, 4096)	16,781,312
Dense	(None, 7)	28,679

**Table 4 jimaging-07-00144-t004:** Accuracy, Macro Average F1 and Micro Average F1 of AlexNet with the 4 different configurations. AlexNet is trained on the conveyor-belt-oriented training set and tested on the conveyor-belt-oriented and on the Littered waste testing sets. Results report the average and the standard deviation of 5 runs with different seeds. Difference in results is not statistically different using the *t*-test.

AlexNet		NotAug	BackRep & BackRep	NotAug	LittleAug
Conveyor-belt-oriented test					
Accuracy		0.683(±0.020)	0.444(±0.017)	0.624(±0.017)	0.697(±0.036)
Macro AVG	F1	0.675(±0.011)	0.429(±0.012)	0.620(±0.018)	0.690(±0.034)
Micro AVG	F1	0.683(±0.009)	0.436(±0.016)	0.626(±0.016)	0.695(±0.035)
Littered Waste test					
Accuracy		0.138(±0.037)	0.161(±0.045)	0.149(±0.045)	0.147(±0.019)
Macro AVG	F1	0.120(±0.045)	0.134(±0.049)	0.129(±0.045)	0.125(±0.022)
Micro AVG	F1	0.110(±0.045)	0.139(±0.045)	0.130(±0.044)	0.112(±0.018)

**Table 5 jimaging-07-00144-t005:** Accuracy, Macro Average F1 and Micro Average F1 of InceptionV4 with the 4 different configurations. InceptionV4 is trained on the conveyor-belt-oriented training set and tested on the conveyor-belt-oriented and on the Littered waste testing sets. Results report the average and the standard deviation of 5 runs with different seeds. The symbols †, ★ and ∘ indicate a statistically significant difference between two results with a 95% of confidence level with the Student’s *t* test.

InceptionV4	NotAug	BackRep & BackRep	NotAug	LittleAug
Conveyor-belt-oriented test				
Accuracy	0.844(±0.009)	0.758(±0.005)	0.847(±0.005)	0.851(±0.008)
Macro AVG F1	0.836(±0.011)	0.738(±0.004)	0.832(±0.008)	0.847(±0.009)
Micro AVG F1	0.843(±0.009)	0.757(±0.005)	0.846(±0.005)	0.851(±0.008)
Littered Waste				
Accuracy	$0.409\left(\pm 0.012\right){\;}^{★†}$	$0.442\left(\pm 0.015\right){\;}^{†}$	$0.488\left(\pm 0.025\right){\;}^{★\circ }$	$0.451\left(\pm 0.014\right){\;}^{\circ }$
Macro AVG F1	$0.427\left(\pm 0.011\right){\;}^{★}$	0.422(±0.016)	$0.483\left(\pm 0.032\right){\;}^{★}$	0.461(±0.020)
Micro AVG F1	$0.414\left(\pm 0.013\right){\;}^{★}$	0.427(±0.016)	$0.482\left(\pm 0.028\right){\;}^{★\circ }$	$0.439\left(\pm 0.015\right){\;}^{\circ }$

## Data Availability

The Littered waste dataset will be shared using ResearchGate.

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
