# Peer review of "Data Augmentation Using Background Replacement for Automated Sorting of Littered Waste"

_2313-433X, 2021, doi:10.3390/jimaging7080144_

Round 1

Reviewer 1 Report

General statement:

While the paper presents an interesting and innovative approach there are some formal aspects that need to be considered. This includes primarily the general structure of the manuscript and the used language, as it is partly written in a very narrative manner including non-scientific language. Especially, the introduction and the conclusion sections are not well elaborated and in need of improvement, as the phrasing is quite informal and the relevance for real-life applications remains unclear. However, the description of different datasets is elaborated well, which is beneficial for those interested in the field but with little prior knowledge.

Language: Proofreading by a native speaker and/or translation agency is highly advised.

Comments as following:

  1. Terminology issue. The term “abandoned waste” is not incorrect but the widely used technical term is “littered waste” or “litter” and the phenomenon is called “littering”. The reviewer highly suggests the use of the term “littered waste”.
  2. The abstract is the visiting card of a scientific paper. Hence, it should (1) not be misleading, (2) include all relevant aspects of the paper, including the main results and the relevance of the work itself.
  3. General structure of manuscript. The manuscript deviates from the classic and proven IMRAD structure, although a division into Introduction, Material and Methods, Results, Discussion and Conclusions would fit perfectly. In fact, that is not a big issue, but some sections are a bit jumbled. For example,
    (1) the aim/goal of the publication and research questions are missing (usually at the end of the introduction). Instead, at this point “…major contributions of the paper…” are presented, in a rather conclusive manner (see lines 52-56);
    (2) and results of conducted work and implementation, which are in fact results, are presented and discussed(!) in the methods section (e.g. lines 166-172).
  4. Lines 1-3 (and lines 26-28). Of course, the problem of littering and littered waste itself cannot rely on augmenting waste treatment or waste sorting plants. Where is the connection? These two issues have absolutely nothing to do with each other.
    Suggestion: Despite the fact that sorting and recycling efficiencies are improved by augmenting modern waste treatment plants … another problem is arising. More and more waste is being littered in the environment instead of being collected properly …
  5. Lines 17-25. There are publications revealing that there is (almost) no part of earth, where micro plastics are not detectable. I would suggest to add that interesting and disturbing fact to the introduction section. Further, information on which types of waste are usually found in the environment in general would be interesting to show later whether your dataset is prepared for these specific tasks.
  6. Line 77. You used the abbreviation CNN for convolutional neural network without explaining them beforehand. Please ensure that the abbreviations are written out when they are used for the first time in the manuscript.
  7. Line 83 – 85. If the aim of the research is to develop waste picking robots or apps for mobile phones, please emphasize how and where this could be implemented in detail and what specific use cases you see for that technology. This information should be part of the introduction.
  8. Lines 93-95 (and 100 and 118ff). Terminology issue. The phrase “waste-in-normal-setting” is absolutely inappropriate as littered waste is everything but normal. Please find another terminology to describe the dataset/testset to use throughout the whole manuscript.
    For example, you could use “littered waste in environment”, or “littered waste testset”.
    This is an interdisciplinary manuscript, hence, it is necessary to use the appropriate technical vocabulary of both disciplines.
  9. Line 105 (and Table 1). Please us thousands separators according to the style of the journal.
  10. Table 1 and Table 2. Although the presented categories are the same in both tables, the order is different. To enable the reader to check easily whether the shares of materials are the same in both datasets change the order of the second Table to match the first one. Further, adding information on why you choose to have these distribution would be interesting. E.g. is there a reason that you did take more images of compost but less of paper?
  11. Lines 116-117. Some readers may not know the difference between the subsets for validation and test. Please explain in more detail.
  12. Line 166. Narrative and not scientific language. Which aspects of BackRep are not perfect? Why is it useful? Please mention examples (for application).
  13. Lines 174-175. The sentence is an example of English language that needs improvement.
  14. Tables 4 and 5. The terms Macro AVG F1 and Micro AVG F1 are not defined. Please define and/or add formula and/or add reference.
  15. Tables 4 and 5. Why are significantly different results only shown for InceptionV4 and not for AlexNet. If the t-test revealed no statistically significant result, please indicate and not conceal it.
  16. Tables 4 and 5. The configuration LittleAug is not well defined in the text. What do the results mean? How can they be interpreted?
  17. Lines 216-218. The phrase “…have important results…” is neither scientifically sound nor meaning anything that is worth mentioning in a paper. Please rephrase.
  18. Lines 220-221. The phrase “…has extremely better performance…” is not scientifically as well. Please rephrase and stick to the facts: What or how much is the difference?
  19. Line 239. The sentence is an example of English language that needs improvement.
  20. Lines 241-244. Please elaborate the conclusion section. Take up the most important points from the introduction, link them with the results of your research and present the open questions/tasks for further research.
  21. Lines 256-330. Please check the structure of the references and consult the guidelines of the journal. There are some inconsistencies, e.g.

Reviewer 2 Report

The most important problem of the paper is its background and relevant references. The main argument of the paper is about data augmentation, but recent papers in the field which are related to the automation of data augmentation are missing. The following papers need to be cited, and it should be explained that how this method can be integrated with them, especially reference [4], which also deals with augmentation for small datasets.  

[1] AutoAugment: Learning Augmentation Policies from Data

[2] Greedy AutoAugment

[3] Fast AutoAugment

[4] Greedy auto-augmentation for n-shot learning using deep neural networks

The second problem of the paper is related to its English, which needs improvements; please proofread and resubmit the paper, these are some examples:

  • were not more adequate.
  • may be have misinterpreted a
  • selected to different state-of-the-art 
  •  the the top dense layer

Round 2

Reviewer 1 Report

Line 4: however

Line 7: waste

Line 10: those

Line 16:  ;

...

It is not the task of a reviewer to find the typos and spelling errors of the authors.